# Development and validation of a nomogram for the prediction of late culture conversion among multi-drug resistant tuberculosis patients in North West Ethiopia: An application of prediction modelling

**Denekew Tenaw Anley**[1]*, **Temesgen Yihunie Akalu**[2], **Mehari Woldemariam Merid**[2], **Anteneh Mengist Dessie**[1], **Melkamu Aderajew Zemene**[1], **Biruk Demissie**[1], **Getachew Arage**[3]

1 Department of Public Health, College of Health Sciences, Debre Tabor University, Debre Tabor, Ethiopia, 2 Department of Epidemiology and Biostatistics, Institute of Public Health, College of Medicine and Health Sciences, University of Gondar, Gondar, Ethiopia, 3 Department of Pediatrics and Child Health Nursing, Debre Tabor University, Debre Tabor, Ethiopia

* denekewtenaw7@gmail.com

## Abstract

### Introduction

Multi-drug resistant tuberculosis has impeded tuberculosis prevention and control due to its low treatment efficiency and prolonged infectious periods. Early culture conversion status has long been used as a predictor of good treatment outcomes and an important infection control metric, as culture-negative patients are less likely to spread tuberculosis. There is also evidence that suggests that delayed sputum conversion is linked to negative outcomes. Therefore, this study was aimed at developing a nomogram to predict the risk of late culture conversion in patients with multi-drug resistant tuberculosis using readily available predictors.

### Objective

The objective of this study was to develop and validate a risk prediction nomogram for the prediction of late culture conversion among multi-drug resistant tuberculosis patients in North-West Ethiopia.

### Methods

Multi-drug resistant tuberculosis data from the University of Gondar and the Debre Markos referral hospitals have been used and a total of 316 patients were involved. The analysis was carried out using STATA version 16 and R version 4.0.5 statistical software. Based on the binomial logistic regression model, a validated simplified risk prediction model (nomogram) was built, and its performance was evaluated by assessing its discriminatory power

**Data Availability Statement:** All relevant data are within the paper and its Supporting Information files.

**Funding:** The authors received no specific funding for this work.

**Competing interests:** The authors have declared that no competing interests exist.

**Abbreviations:** AUC, Area Under Curve; BMI, Body Mass Index; DCA, Decision Curve Analysis; DM, Diabetes Mellitus; DR-TB, Drug-Resistant Tuberculosis; HIV, Human Immunodeficiency Virus; IQR, Inter Quartile Range; IRB, Institutional Review Board; MDR-TB, Multi-Drug Resistant Tuberculosis; NPV, Negative Predictive Value; PPV, Positive Predictive Value; TB, Tuberculosis; TIC, Treatment Initiating Center; UoGCSH, University of Gondar Compressive Specialized Hospital; WHO, World Health Organization.

and calibration. Finally, decision curve analysis (DCA) was used to assess the generated model's clinical and public health impact.

## Results

Registration group, HIV co-infection, baseline BMI, baseline sputum smear grade, and radiological abnormalities were prognostic determinants used in the construction of the nomogram. The model has a discriminating power of 0.725 (95% CI: 0.669, 0.781) and a P-value of 0.665 in the calibration test. It was internally validated using the bootstrapping method, and it was found to perform similarly to the model developed on the entire dataset. The decision curve analysis revealed that the model has better clinical and public health impact than other strategies specified.

## Conclusion

The developed nomogram, which has a satisfactory level of accuracy and good calibration, can be utilized to predict late culture conversion in MDR-TB patients. The model has been found to be useful in clinical practice and is clinically interpretable.

## 1. Introduction

Multidrug-resistant (MDR) tuberculosis (TB), TB that is resistant to the two most effective first-line anti-TB drugs isoniazid and rifampicin, is an increasing global problem and a major burden for some developing countries [1]. Worldwide in 2019, 3.3% of new TB cases and 17.7% of previously treated cases had MDR/Rifampicin Resistant Tuberculosis (RR-TB). Overall, an estimated 465 000 new cases of MDR-TB occurred and about 182 000 people died of MDR-TB in 2019. Ethiopia is also among the 30 high burden TB, TB/HIV and MDR-TB countries that accounts for 87% of all global cases [2]. There is a commitment to end the TB epidemic through adopting End TB Strategy and the Sustainable Development Goals (SDGs) [3]. However, MDR-TB hampered the prevention and control of tuberculosis due to the poor effectiveness of the treatment and longer infectious periods of MDR-TB [4].

The treatment of MDR-TB is lengthy (nine to 24 months), expensive and toxic regimens, complicated by potentially severe adverse effects, and is more costly than treatment of drug susceptible TB [5, 6]. In addition, treatment outcomes are sub-optimal, with only 57% of MDR-TB patients successfully treated globally in 2019 [2]. Thus, monitoring treatment responses is essential to control MDR-TB and prevent the emergence of extensively drug-resistant (XDR-TB) strains that arise from the mismanagement of MDR-TB individuals [7]. Culture conversion, defined as two consecutive negative sputum cultures taken at least 30 days apart following an initial positive culture, is an interim monitoring tool for MDR-TB treatment [8].

The status of early culture conversion has been widely used as an indicator of favorable treatment outcome and an important infection control measure, as culture-negative patients are less likely to transmit TB [8–11]. There is also evidence that indicates delayed sputum conversion is associated with unfavorable outcomes, more specifically with failure and death [12, 13]. Overall, median time to sputum culture conversion among patients with treatment success was significantly shorter than in those who had poor outcomes (2 months versus 7 months). Similarly, a study in china indicates that the median sputum culture conversion time of

patients with successful treatment outcomes was 92 days which was much shorter than 174 days for the patients with treatment failure or death [12, 13].

Several factors will influence the time to sputum culture conversion of patients treated for MDR-TB. The most common factors are: alcohol drinking [9, 14, 15], cavitary TB [10, 14, 16–18], sputum smear grading [9, 14, 15, 19], smoking [15, 18, 19], body mass index (BMI) [18, 20, 21], type of resistance [9, 10, 15, 16, 18], and HIV status [16, 20]. Although too many studies have investigated factors associated with delayed culture conversion among patients treated for MDR-TB, risk scores have not yet been developed to predict this delayed culture conversion. Developing a clinical risk score using readily available predictors to identify individuals at increased risk of delayed culture conversion could be a useful aid in clinical decision making. Therefore, this study is aimed at developing a nomogram to predict the risk of late culture conversion in patients with MDR-TB using readily available predictors.

## 2. Materials and methods

### 2.1. Study design and setting

A multi-center retrospective follow-up study design was undertaken in two Treatment Initiating Centers (TICs) in North West Ethiopia from September 2010 to July 2020. The University of Gondar Compressive Specialized Hospital (UoGCSH) was the first TIC. It is 737 kilometers from Ethiopia's capital city, Addis Ababa. The hospital is one of the region's largest tertiary-level teaching and referral facilities. It has a capacity of more than 500 beds. The hospital serves a population of 5 million people in the North Gondar administrative zone and the surrounding region as a referral facility. It has been treating MDR-TB patients since September 2010, and approximately 450 MDR-TB patients have been enrolled since therapy began. In 2018, the hospital began delivering the novel short-term MDR-TB treatment regimen, and to date, more than fourteen patients have been treated.

Debre Markos Referral Hospital, located 300 kilometers from Addis Ababa, was the second TIC. The hospital serves a population of more than 3.5 million people. It contains 140 beds and 152 employees who provide inpatient and outpatient care. The hospital has a TB/HIV clinic as well as an MDR-TB ward for MDR-TB patients' diagnosis and treatment. In 2015, the hospital began offering MDR-TB testing and treatment. Since then, it has enrolled 95 MDR-TB patients and received 31 transferred patients, bringing the total number of MDR-TB patients served to 126.

### 2.2. Population

All MDR-TB patients enrolled in North West Ethiopia represented as the source population, while those enrolled in the two TICs were the study population. The study included all MDR-TB patients who had follow-ups at the UoGCSH and Debre Markos referral hospital. The study excluded individuals with Extrapulmonary Tuberculosis (EPTB) and those for whom the date of culture conversion couldn't be verified.

### 2.3.Variables of the study

The outcome variable is late culture conversion (yes/no). Culture conversion at month two is widely used as a proxy measure of the success of tuberculosis (TB) chemotherapy [22, 23]. Late culture conversion was defined as sputum culture conversion that occurred after two months of treatment initiation. Prognostic determinants were; Sex, Age, Residence, Treatment supporter, Registration group, Functional status, Radiological finding, HIV co-infection,

Fluoroquinolone resistance, Baseline anemia, Regimen type, Body Mass Index, Baseline sputum smear grade and Major adverse event.

## 2.4. Data collection procedure and quality control

Using different literatures and piece of evidence available on the medical charts of patients, a structured data extracting tool (checklist) was built. Prognostic determinants of late culture conversion such as sociodemographic factors, treatment-related factors, laboratory-related factors, comorbidities, and behavioral factors were extracted. Body Mass Index (BMI) of patients was calculated by taking the baseline weight in kilogram and dividing it by height in meters squared. The data gathered, were checked for completeness and accuracy consistently in daily bases.

## 2.5. Ethics approval and consent to participate

The Institutional Review Board (IRB) of the University of Gondar granted ethical approval with the reference number /IPH/1440/2021. In all Treatment Initiating Centers (TICs), a permission letter was acquired from the hospital administration and the DR-TB ward focal person. To maintain anonymity, personal identifiers were not included. Because this is solely secondary data, informed consent was not obtained directly from the study participants. Hence, the university's institutional Review Board (IRB) waived the study participants' permission.

## 2.6. Data processing and analysis

The coded data containing prognostic determinants were entered into Epi-info version 7. Then it was exported to STATA version 16 and R version 4.0.5 statistical software for analysis. Descriptive statistics, frequencies, and percentages were done for categorical variables. The normality distribution test was done using the Kolmogorov-Smirnov test. Normally distributed continuous variables were summarized using mean and standard deviation. Variables with normality assumption failed, were summarized using median and Inter-quartile range (IQR). Variables were also checked for possible outlier and multicollinearity using pseudo linear regression.

## 2.7. Missing data handling

**Variables with missing values were**; Residence 30 (9.5%), Baseline sputum culture grade 22 (6.96%), Functional status 20 (6.3%), and baseline anemia status 11 (3.5%). The variable income was dropped because it was missed for majority of observations 268 (84.8%). Multiple imputation technique using "**mice**" package in r and random Forest method was used.

**2.7.1. Model development.** Potential variables for late culture conversion prediction model development were considered based on their easily obtainability, biologically plausible relationship with the outcome, and ease of interpretation in clinical practice. Binomial logistic regression model was used to develop the risk prediction model. Univariable analysis was undertaken to select prognostic determinants of late culture conversion. Those variables with p-value of 0.25 and below in univariable analysis were entered into multivariable analysis. The statistically significant association was declared at p-value of 0.05 and below. The final simplified multivariable logistic regression model developed was presented in the form of risk prediction nomogram.

**2.7.2. Model performance evaluation.** The resulting nomogram's prediction capacity was evaluated in terms of discriminatory power and calibration. Calculating c-statistics in roc-

curve analysis was used to measure the discriminatory strength of the produced nomogram. The c-statistics could range from 0.5 (no prediction capacity) to 1 (high predictive ability) (perfect discrimination) [24, 25]. It's a metric for how well the model distinguishes between patients who had a positive outcome and those who hadn't. The calibration plot and Hosmer-Lemeshow test were used to visualize the model's calibration performance. A P-value>0.05 in the model calibration test (Hosmer-Lemeshow test) indicated acceptable model calibration. The usual performance measurements for risk prediction models (discrimination and calibration) were unable to address the question of how beneficial the produced nomogram would be in clinical practice. As a result, decision curve analysis (DCA) was done with the intention of developing a more clinically interpretable risk prediction model.

**2.7.3. Model validation.** The model was internally validated and bootstrapping method was used for the purpose of model validation. This method was chosen above other methods of model validation because it allows for the development of the most stable prediction model in a limited number of research participants. The bootstrapping procedure was made in R version 4.0.5. The random sampling was made by 10,000 reputations of the original set with replacement. The 95% confidence probablity was set and, the output which contains the original coefficients, the bias and standard error (SE) was found.

The study was reported per the transparent reporting of a multivariable prediction model for individual prognosis or diagnosis (TRIPOD) statement [26].

# 3. Results

A total of 578 patients were enrolled and started MDR-TB treatment in the two hospitals in North West Ethiopia from September 2010 to April 2021. The analysis was carried out on the total of 316 patients for whom culture conversion status was registered. The procedure of selecting participants is depicted in the diagram below (**Fig 1**).

## 3.1. Socio-demographic characteristics

Among the participants, 200 (63.3%) were men. The median age of the participants was 30 years old, with a 17-years interquartile range (IQR). In terms of educational attainment, 121 (38.3%) of the participants had no formal schooling (**Table 1**).

## 3.2. Behavioral characteristics

About 10.76% of study subjects had a history of cigarette smoking and 30.06% of participants had history of alcohol use (**Fig 2**).

## 3.3. Clinical characteristics

Anemia affected 46.2% of the patients, and the majority (73.42%) had a low BMI. About 77.22% of the subjects had previous history of anti-TB treatment. Comorbid illnesses were diagnosed in one-third of the individuals, and 22.8% were HIV positive (**Table 2**).

## 3.4. Late culture conversion

The cumulative incidence of late culture conversion was found to be 44.6% (95% CI: 39.2, 50.2). A total of 175 (55.38%) MDR-TB patients had culture conversion at two months of anti-MDR-TB treatment initiation.

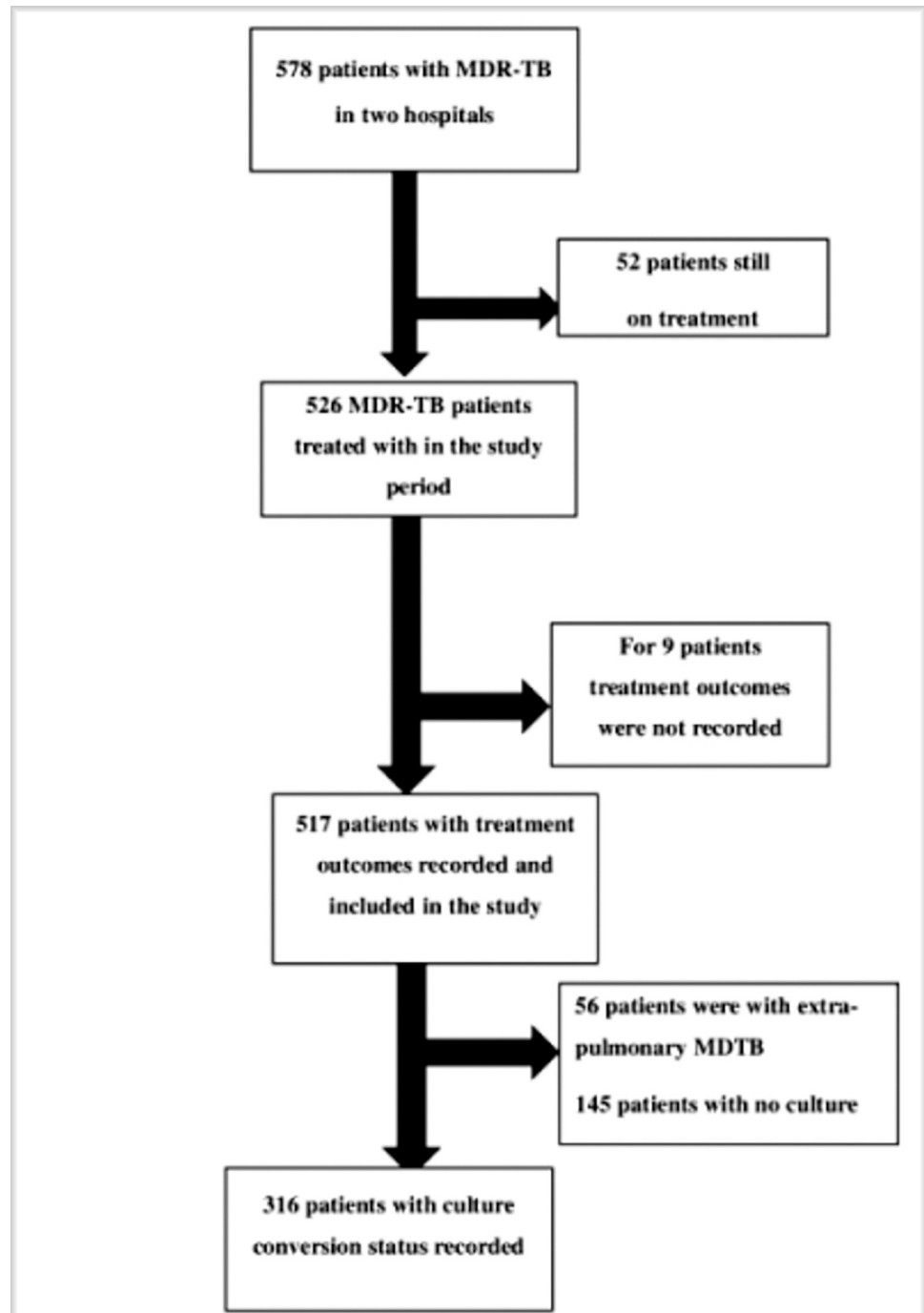

**Fig 1. Flowchart of participant selection for the development of nomogram for prediction of late culture conversion among MDR-TB patients, and reasons for exclusion, North West Ethiopia, September 2010 to July 2020.**

## 3.5. Development of an individualized risk prediction model

Binomial logistic regression analysis was used to create a customized risk prediction model. In univariable analysis, fourteen candidate variables were fitted. To avoid the

**Table 1. Socio-demographic characteristics of MDR-TB patients in North West Ethiopia, 2010 to 2020 (N = 316).**

| Characteristics | Frequency (%) |
|---|---|
| **Gender** | |
| Male | 200 (63.3) |
| Female | 116 (36.7) |
| **Age** (Median ± IQR) | 30±17 |
| **Occupation** | |
| Government employee | 24 (7.6) |
| Self-employed | 45 (14.2) |
| Farmer | 101 (32) |
| Unemployed | 6 (2) |
| Student | 36 (11.4) |
| Daily laborer | 60 (19) |
| House wife | 31 (9.8) |
| Others* | 13 (4) |
| **Educational status** | |
| No formal education | 121 (38.3) |
| Primary | 105 (33.2) |
| Secondary | 63 (20) |
| Tertiary | 27 (8.5) |
| **Marital status** | |
| Single | 102 (32.28) |
| Married | 130 (41.14) |
| Divorced | 54 (17.09) |
| Widowed | 9 (2.85) |
| Others** | 21 (6.64) |
| **Religion** | |
| Orthodox | 292 (92.41) |
| Muslim | 21 (6.65) |
| Others*** | 3 (0.95) |
| **Residence** | |
| Rural | 157 (49.68) |
| Urban | 159 (50.32) |
| **Treatment supporter** | |
| Yes | 268 (84.81) |
| No | 48 (15.19) |

**Note:** Others* = preschool children, Others** = children bellow 18 years, Others*** = protestant and catholic

problem of overfitting in prediction analytics, the number of candidate variables was limited to 14. The number of events per parameters (EPP) has to be at least greater or equal to 10 to prevent the problem of over fitting. Hence, 14 variables could insure this for the total of 145 events (late culture conversion). The majority of these variables are easily detectable at the time of the patient's enrolment. These variables were; Sex, Age, Residence, Treatment supporter, Registration group, Functional status, Radiological finding, HIV co-infection, Fluoroquinolone resistance, Baseline anemia, Regimen type, Body Mass Index (BMI), Baseline sputum smear grade and Major adverse event (MAE) (**Table 3**).

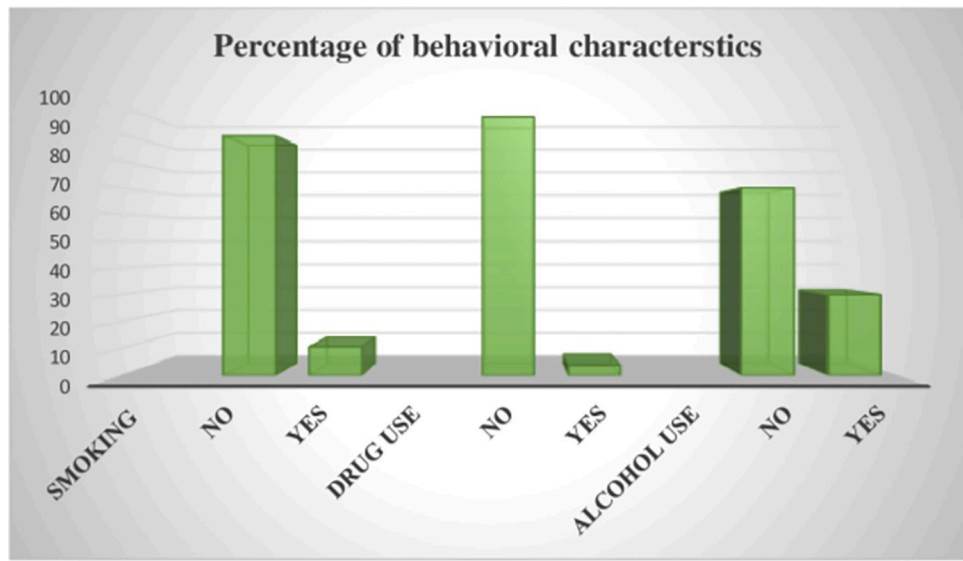

**Fig 2. Behavioral characteristics of MDR-TB patients in North West Ethiopia, September 2010 to July 2020.**

## 3.6. Nomogram of the final model

The nomogram was constructed using the predictors which were found to have statistically significant association with the outcome (late culture conversion). These predictors were; Registration group of the patient (New/Previously treated), HIV co-infection (Yes/No), Baseline BMI ($<18.5/ \geq18.5$), sputum smear grade ($<3+/3+$) and radiological finding (non-cavitary lesions/ Cavitary lesions). The developed nomogram can be used to calculate the risk of late culture conversion for individual patients, using which they can be classified as at lower or higher risk of late culture conversion in their course of treatment (**Fig 3**).

## 3.7. Performance of the nomogram developed

Discriminatory power and calibration are the two metrics used to assess the performance of risk prediction models. As a result, the nomogram's performance was assessed by looking at its calibration and discriminatory power. The AUC of the receiver operating characteristics curve (ROC-curve) was used to determine its discriminating power and it was found to be (AUC = 0.725, 95% CI; 0.669, 0.781) (**Fig 4**).

Estimated risk of late culture conversion = $1/(1 + \exp - (-0.323 + 0.63\times$ BMI($<18.5$) $+ 0.87 \times$ smear grade($>3+$) $+ 0.97\times$ registration group (previously treated) $+ 0.67 \times$radiological finding (cavitory lesions) $+ 0.7\times$ HIV co-infection (Yes). The equation is actually the probability equation of the binomial logistic regression model. This is mathematics intensive and not feasible to calculate. Hence, the developed nomogram is a replace for this sophisticated equation to calculate the probability or risk of late culture conversion. The nomogram developed doesn't require any calculator, for it is simple and user friendly graphical interface to calculate the individualized risk of patients for the outcome of interest.

The model calibration was assessed by plotting the actual probability against the predicted one. It was assessed using both the calibration plot and the goodness of fit test (Hosmer-Lemeshow test) (**Fig 5**).

**Table 2. Clinical characteristics of MDR TB patients in Northwest Ethiopia, 2010 to 2020 (N = 316).**

| Characteristics | Frequency (%) |
|---|---|
| **Type of drug regimen** | |
| Long term regimen | 290 (91.77) |
| Short term regimen | 26 (8.23) |
| **Registration group** | |
| Previously treated | 244 (77.22) |
| New | 72 (22.78) |
| **Model of care** | |
| Hospitalized | 294 (93.04) |
| Ambulatory | 22 (6.96) |
| **Comorbidity** | |
| No | 222 (70.25) |
| Yes | 94 (29.75) |
| **HIV co-infection** | |
| No | 244 (77.22) |
| Yes | 72 (22.78) |
| **Major adverse event** | |
| No | 266 (84.18) |
| Yes | 50 (15.82) |
| **Complications** | |
| No | 200 (63.29) |
| Yes | 116 (36.71) |
| **Type of resistance** | |
| Mono | 183 (57.91) |
| MDR | 105 (33.23) |
| Poly | 28 (8.86) |
| **Fluoroquinolone resistance** | |
| No | 311 (98.42) |
| Yes | 5 (1.58) |
| **Baseline sputum smear grade** | |
| <3+ | 197 (62.34) |
| > = 3+ | 119 (37.66) |
| **Baseline anemia** | |
| No | 170 (53.8) |
| Yes | 146 (46.2) |
| **Radiological abnormalities** | |
| Cavitary lesions | 106 (33.54) |
| Non-cavitary lesions | 210 (66.46) |
| **Body mass index** | |
| <18.5kg/m$^2$ | 232 (73.42) |
| ≥18.5kg/m$^2$ | 84 (26.58) |

## 3.8. The classification of risk

The nomogram-based risk probability calculation for individual patients is far too simple for any health practitioner at any level to execute. Patients are categorized as low risk (risk probability <0.4562) or high risk (risk probability ≥0.4562) of late culture conversion using the optimum cutoff point (0.4562) found by the Youden index method. About 44.9% and 55.1% of patients were found to be at the low and high risk of late culture conversion, respectively. Late

**Table 3. Univariable and multivariable logistic regression analysis using potential predictors of late culture conversion in patients with multidrug-resistant tuberculosis, in North West Ethiopia, 2010–2020 (N = 316).**

| Predictors selected by lasso algorithm | Univariable analysis | | Multivariable analysis | |
|---|---|---|---|---|
| **Sex** | **Coef [95% CI]** | **p-value** | **Coef [95% CI]** | **p-value** |
| Male | 0 | | | 0.069 |
| Female | 0.29 [-0.171, 0.748] | 0.22 | 0.489[-0.038, 1.016] | |
| **Age** | | | | 0.094 |
| <45 years | 0 | | | |
| ≥45 years | 0.81[0.207, 0.1.407] | 0.008 | 0.57[-0.099, 1.245] | |
| **Residence** | | | | |
| Urban | 0 | | | |
| Rural | 0.09[-0.344, 0.543] | 0.660 | — | — |
| **Treatment supporter** | | | | |
| No | 0 | | | |
| Yes | -0.35 [-0.978, 0.286] | 0.283 | — | — |
| **Functional status at admission** | | | | |
| Ambulatory | 0 | | | 0.102 |
| Bedridden | 0.94[0.143, 1.744] | 0.021 | 0.77[-0.155, 1.699] | |
| **Registration group@** | | | | |
| New | 0 | | | 0.002** |
| Previously treated | 0.94[0.366, 1.513] | 0.001 | 0.97 [0.344, 1.597] | |
| **HIV co-infection@** | | | | |
| No | 0 | | | 0.020* |
| Yes | 0.72 [0.184, 1.252] | 0.008 | 0.70 [0.108, 1.296] | |
| **Fluoroquinolone resistance** | | | | |
| No | 0 | | | 0.069 |
| Yes | 1.63 [-0.577, 3.828] | 0.148 | 1.66 [-0.735, 4.052] | |
| **Body mass index@** | | | | |
| ≥18.5 | 0 | | | 0.033* |
| <18.5 | 0.57[0.533, 1.091] | 0.031 | 0.63 [0.050, 1.205] | |
| **Baseline anemia** | | | | |
| No | 0 | | | 0.364 |
| Yes | 0.30 [-0.144, 0.748] | 0.184 | 0.25 [-0.174, 0.284] | |
| **Baseline smear grade@** | | | | 0.001** |
| <3+ | 0 | | | |
| ≥3+ | 1.10[0.631, 1.573] | 0.000 | 0.87 [0.344, 1.400] | |
| **Radiological finding@** | | | | |
| Non-cavitary | 0 | | | 0.012* |
| Cavitary lesions | 0.91[0.429, 1.386] | 0.000 | 0.669 [0.147, 1.193] | |
| **Regimen type** | | | | |
| Short term regimen | 0 | | | |
| Long term regimen | 0.10[-0.709, 0.914] | 0.923 | — | — |
| **Major adverse event** | | | | |
| No | 0 | | | — |
| Yes | -0.03 [-0.638, 0.579] | 0.923 | — | |
| **Intercept** | -0.323[-.604, -0.042] | 0.024 | | |

Coef. = coefficients, CI = confidence interval, **@** = Variables included in the final simplified model.

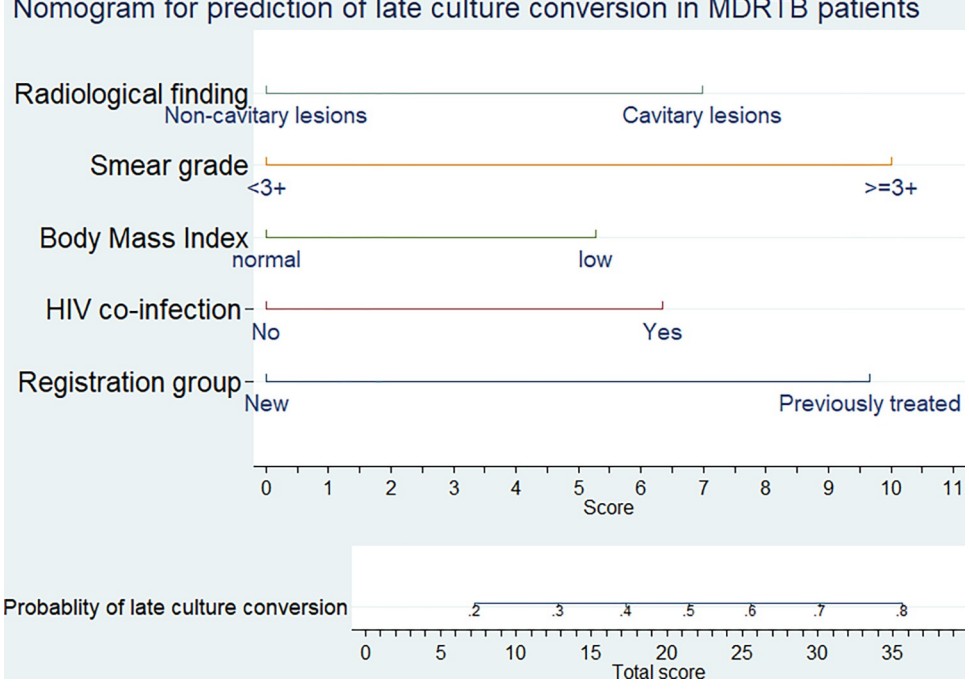

**Fig 3. Nomogram for the prediction of late culture conversion among MDR-TB patients in North West Ethiopia, September 2010 to July 2020.**

culture conversion was found to be 63.8% and 21.1% among patients at high and low risk of the outcome, respectively (**Table 4**).

## 3.9. performance of the nomogram at several cut-off points

The optimum cut-off points were determined by methods like Youden index, MaxEfficiency, MaxProdSpSe and SpEqualSe. The cut-off point identified by the first three methods (Youden index, MaxEfficiency and MaxProdSpSe) was found to be 0.4562 and the one identified by SpEqualSe method was 0.4955. The following table illustrates the performance of the nomogram at the above cut-off points and other selected cut-off points and it is described in terms of sensitivity, specificity, PPV and NPV (**Table 5**).

## 3.10. Model validation

Internally, the model was validated using the bootstrapping method. It was found to have equivalent discriminatory power to the original model when bootstrap samples of 10,000 repeats with replacement were drawn. The large number of repetitions (10,000 with replacement) was chosen in order to construct a more stable prediction model (**Fig 6**).

## 3.11. Decision curve analysis (DCA)

The red line represents the developed risk prediction nomogram, the thin black line represents the assumption that all patients are at risk of late culture conversion and the thick black line represents the assumption that none of the patients are at risk of late culture conversion. The y-axis of the curve represents the standard net benefit of using either the developed model or the two extreme approaches (all or none approaches) and the x-axis represents different threshold probabilities with possible cost benefit ratio. Hence, the curve generally shows the

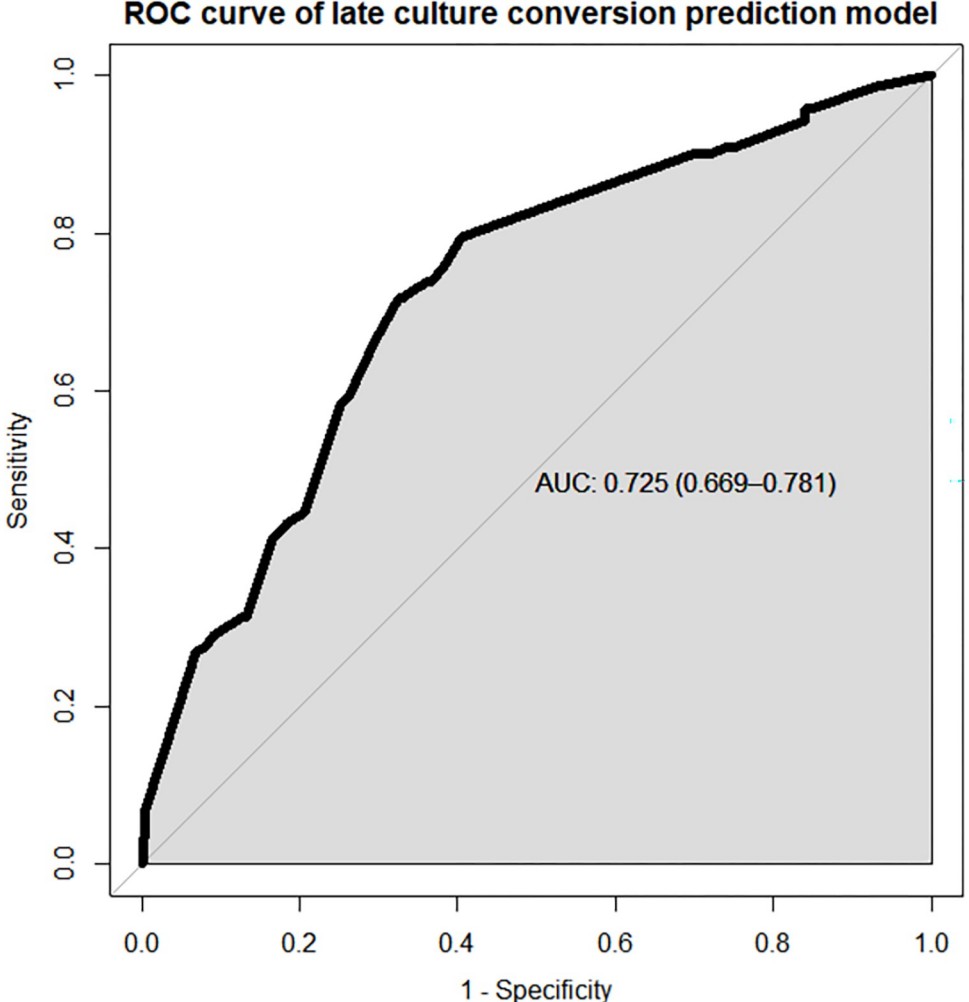

**Fig 4. ROC curve of the risk prediction model for prediction of late culture conversion among MDR-TB patients in North West Ethiopia.**

standard net benefit of using the model in comparison to the other two approaches across different threshold probabilities (**Fig 7**).

## 4. Discussion

This study was aimed for determining the extent of late culture conversion and developing a risk prediction nomogram for prediction of late culture conversion among MDR-TB patients. The proportion of late culture conversion was found to be 44.6% (95% CI: 39.2, 50.2). This is an excessively high percentage, as over half of the patients had faced late culture conversion. This finding is almost comparable with the finding of the study conducted in Pakistan (46.6%) [10], but much higher than the finding of a study conducted in South and Southwestern Ethiopia (20%) [27]. This might be because of variation in the status of the setting and other patient characteristics. In fact, this extent of the outcome is lower than the finding of a study conducted in Uganda (55%) [23].

Among 14 candidate prognostic determinants, five of them were found to be significant predictors of late culture conversion. These were; registration group (new/previously treated),

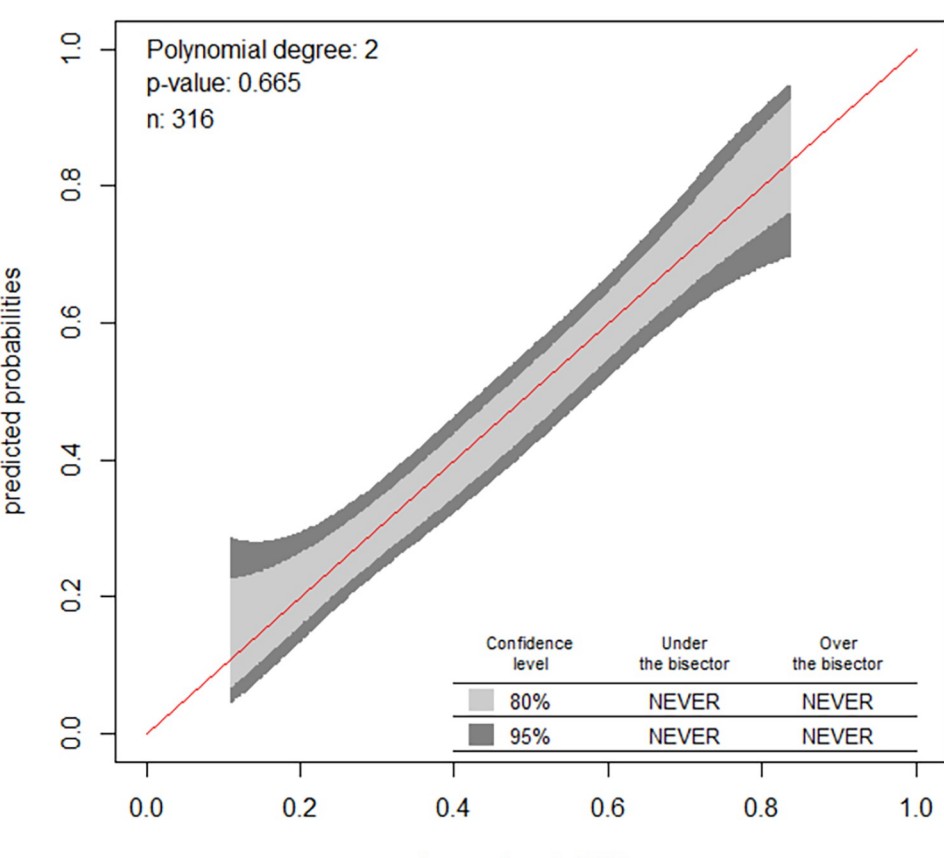

**Fig 5. Observed versus predicted probabilities of late culture conversion in a sample.**

HIV co-infection (yes/no), BMI ($<18.5$/$\geq18.5$), baseline sputum smear grade ($<3+$/$\geq3+$) and radiological findings (cavitary lesions/ no cavitary lesions). This finding on significant predictors of late culture conversion was also supported by different studies conducted across the world [9, 10, 16, 23, 27–29].

For the individualized prediction of late culture conversion among MDR-TB patients, a multivariable risk prediction nomogram was constructed and internally validated. The model was developed with the intent of providing a clinical tool that will help practitioners manage and treat MDR-TB, which is a major challenge in many high-burden countries, especially during current period of global crisis brought on by COVID-19. To make the model constructed

**Table 4. Risk classification of late culture conversion using a nomogram (n = 316).**

| Risk category* | Late culture conversion prediction nomogram | |
|---|---|---|
| | Number of patients (%) | Proportion of late culture conversion |
| Low ($<0.4562$) | 142(44.9%) | 30 (21.1%) |
| High ($\geq0.4562$) | 174 55.1%) | 111(63.8%) |
| Total | 316 (100%) | 141(44.62%) |

* = the risk probability calculated using the nomogram

**Table 5. Performance of the nomogram at different cut-off points.**

| Cut-off points* | Sensitivity | specificity | PPV | NPV |
|---|---|---|---|---|
| 0.4452 | 71.6% | 67.4% | 63.9% | 74.7% |
| **0.4562** | **73.9%** | **63.9%** | **47.2%** | **84.9%** |
| 0.4955 | 66.7% | 70.3% | 64.4% | 72.4% |
| 0.5213 | 70.9% | 60.9% | 46.2% | 84.1% |

* = risk probabilities, PPV = positive predictive value, NPV = negative

more user-friendly during its use in clinical settings, it was chosen to visualize the risk prediction model graphically using a nomogram. The nomogram was formed by combining the above listed easily obtainable prognostic determinants of individuals with MDR-TB. The predictors included in the model were chosen after fitting univariable and multivariable logit based binomial models to see how they related to the outcome. The number of putative prognostic determinants was limited to 14 in order to keep the number of parameters per event to

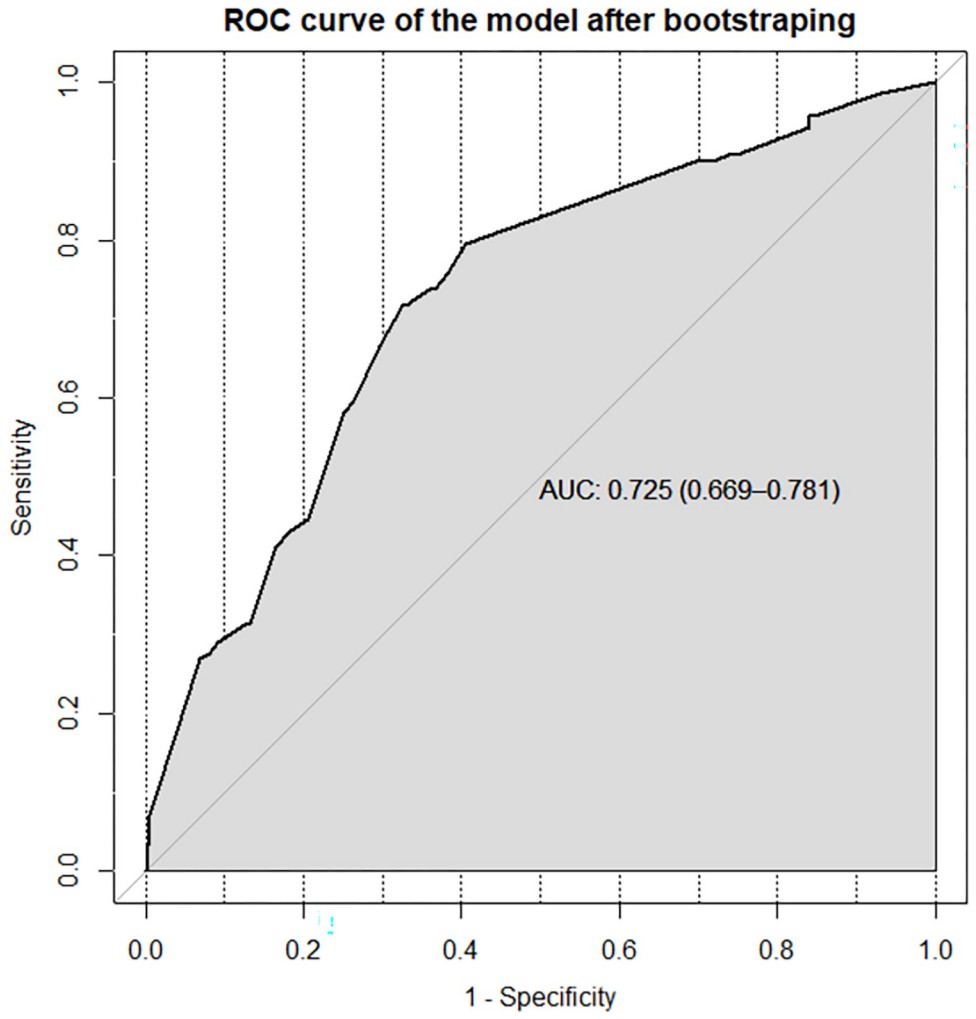

**Fig 6. ROC curve of the model after internal validation using the bootstrapping method.**

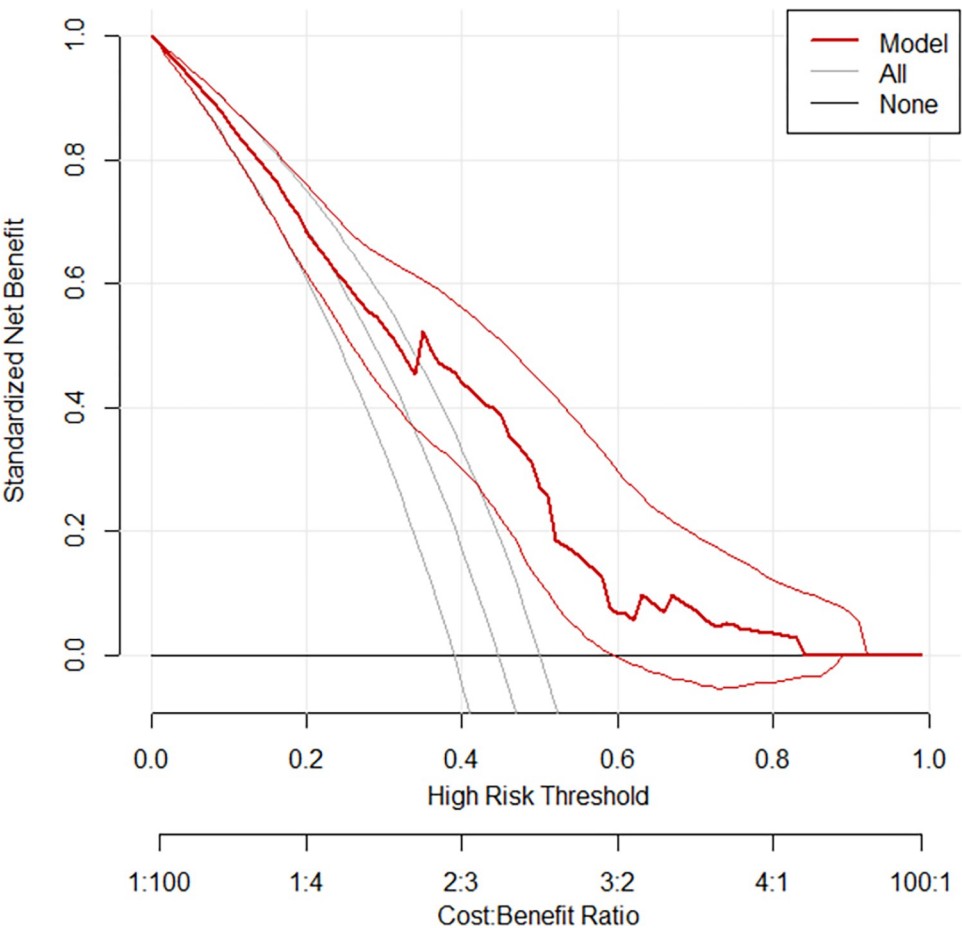

**Fig 7. Decision curve plot showing the net benefit of the developed model for carrying out an intervention measure in MDR-TB patients at risk of late culture conversion.**

ten, allowing for the development of a parsimonious and potentially robust model by avoiding the problem of overfitting in predictive analytics.

The entire dataset was used to build the prediction model, which included the aforementioned predictors. The model was shown to have adequate calibration (p value = 0.665) and a satisfactory level of discrimination accuracy (AUC = 0.725, 95 percent CI; 0.669, 0.781). The performance of our model was shown to be better to that of existing prediction models developed in Brazil to predict poor treatment outcomes in MDR-TB patients which was a nomogram to predict death and dropouts with prediction accuracy of 0.65 and 0.7 respectively [30]. However, when compared to the risk prediction models built to predict poor treatment outcomes among TB patients in Tamaulipas, Mexico (c-statistics = 0.77) [31], and in India (AUC = 0.78) [32], our model performs slightly lower. The identified slight discrepancy in the performance of our model when compared to other prediction models is attributed to the variation in the nature of the outcome of interest and the number of parameters involved in the development of the prediction model.

The model's calibration was evaluated in two methods. The first was by creating a calibration plot. In the calibration plot, the calibration belt was aligned with the y = x line (the 45-degree line) and lies within the 95% confidence range, indicating that the expected and observed probability of late culture conversion are similar. The Hosmer-Lemeshow goodness

of fit test was the second method and the identified insignificant p-value of 0.665, shows that the two probabilities are identical. Model calibration assessment procedures in both ways demonstrated that the model accurately represented the data.

The developed nomogram could be used to calculate the risk probability of late culture conversion for individual patients. Because it is a graphical representation, it is too easy to calculate the risks without even need a calculator, and hence any clinician at any level can do it. Once, the probabilities of late culture conversion determined by the model, patients could be classified as at lower and higher risk of late culture conversion. The Youden Index method, one of the approaches used to establish optimum cut-off points in predictive analytics, identified this point of classification to be 0.4562. If a patient's risk probability, as determined by the created nomogram, is less than 0.4562, he or she is regarded to be at a lesser risk of late culture conversion. Patients with an estimated risk of 0.4562 will be regarded as having a higher chance of late culture conversion. The proportion of late culture conversion among patients at higher risk of the outcome was found to be 63.8% much higher than it was among those who were in lower risk (21.1%). This indicates the strength of the constructed model. Because the predictors included in the model can be available at the time of patient enrolment, it is possible to take the appropriate precautions before the event occurs.

For different cut-off points, the nomogram's performance was assessed in terms of sensitivity, specificity, PPV, and NPV. For instance, at the cut-off point of 0.4562, the model's sensitivity and specificity were found to be 73.9% and 63.9%, respectively. By adjusting the cut-off points, the model's sensitivity and specificity can be enhanced or decreased based on the goal and resources available.

The bootstrap resampling approach of 10,000 repeats with replacement was used to internally validate the model. This approach of model validation was selected over others such as cross-validation and split-half methods because it is essential to construct more robust prediction models, especially when the sample size is small. The model was trained in the bootstrap sample and tested in the original dataset during internal validation using the bootstrapping method. Following internal validation, performance measures revealed a well calibrated model with adequate discrimination power (AUC = 0.725), similar with the original model pointing the fact that the model is not sample dependent, and hence can possibly be applied in external settings.

Decision curve analysis (DCA) was used to show the benefit that the generated nomogram would bring to clinical practice. DCA, the most recent metric for analyzing the real benefit of the prediction model over the treat-all or treat-none scheme, would provide answers to questions that previous performance measurements (discrimination and calibration) could not answer. The nomogram's net benefit was calculated for various threshold probabilities. As it is illustrated by the decision curve plotted, the model has no higher net benefit than treat all scheme across a range of threshold probabilities lower than 0.25. When the patient's threshold probability is greater than 25%, it has a greater net benefit than using the treat all or treat none strategies and it is the case for majority of threshold probabilities listed in the x-axis of the curve. for example, if a patient's personal threshold probability is 50%, the standardized net benefit of using the model is roughly 0.45, with extra benefit over action on all or none of the patients.

Generally, the model was built using fairly ascertainable predictors and presented in the form of a nomogram, making it simple to use yet extremely significant. In a simple manner, a clinician can quantify the probability of late culture conversion and classify patients as having a higher or lower chance of the outcome. Because the nomogram devised does not require any mathematics-intensive calculations, it can be used by any physician at any level. Because it was supported by decision curve analysis, it is also a clinically interpretable model. As to the best of

our knowledge, it is the first prediction model done on late culture conversion among MDR-TB patients.

Therefore, the model will be a useful clinical tool for health care practitioners to apply in their decision-making for individualized management of MDR-TB patients. Furthermore, this study would help with the global End TB program. It can also be used by policymakers and program managers to create patient-specific policies and programs aimed at lowering the high rate of late culture conversion in MDR-TB patients. It would also be used for academic purpose and as reference for researchers. However, the study had a flaw; it would have been better if it had been done with a prospective study design and externally validated.

## 5. Conclusion

The model, which has a satisfactory level of accuracy and good calibration, can be utilized to predict late culture conversion in MDR-TB patients. The model has been found to be useful in clinical practice as it was insured by decision curve analysis.

## Supporting information

**S1 File. The minimal anonymized dataset.**
(DTA)

## Acknowledgments

We are grateful to the University of Gondar Compressive Specialized Hospital and Debre Markos Referral Hospital for their permission and encouragement. We would also like to express our gratitude to the data collectors and card room staff who helped us complete this study.

## Author Contributions

**Conceptualization:** Denekew Tenaw Anley, Temesgen Yihunie Akalu, Mehari Woldemariam Merid, Anteneh Mengist Dessie, Melkamu Aderajew Zemene, Getachew Arage.

**Data curation:** Biruk Demissie.

**Formal analysis:** Denekew Tenaw Anley, Temesgen Yihunie Akalu, Mehari Woldemariam Merid.

**Investigation:** Anteneh Mengist Dessie, Melkamu Aderajew Zemene, Getachew Arage.

**Methodology:** Denekew Tenaw Anley, Temesgen Yihunie Akalu, Mehari Woldemariam Merid, Melkamu Aderajew Zemene.

**Software:** Denekew Tenaw Anley, Mehari Woldemariam Merid, Anteneh Mengist Dessie.

**Supervision:** Temesgen Yihunie Akalu.

**Validation:** Denekew Tenaw Anley.

**Visualization:** Denekew Tenaw Anley, Anteneh Mengist Dessie, Biruk Demissie, Getachew Arage.

**Writing – original draft:** Denekew Tenaw Anley, Biruk Demissie.

**Writing – review & editing:** Melkamu Aderajew Zemene, Getachew Arage.

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
