## [Decision Letter · Decision Letter 0]

6 Jun 2022

PONE-D-22-02882Development and validation of a nomogram for the prediction of late culture conversion among multi-drug resistant tuberculosis patients in North West Ethiopia: an application of Prediction modellingPLOS ONE

Dear Dr. Anley,

Thank you for submitting your manuscript to PLOS ONE. After careful consideration, we feel that it has merit but does not fully meet PLOS ONE’s publication criteria as it currently stands. Therefore, we invite you to submit a revised version of the manuscript that addresses the points raised during the review process.

We look forward to receiving your revised manuscript.

Kind regards,

Dr Aleksandra Barac

Academic Editor

PLOS ONE

Journal Requirements:

a) Did participants provide their written or verbal informed consent to participate in this study?

5. Please include your tables as part of your main manuscript and remove the individual files. Please note that supplementary tables (should remain/ be uploaded) as separate "supporting information" files.

Reviewers' comments:

Reviewer's Responses to Questions

**Comments to the Author**

1. Is the manuscript technically sound, and do the data support the conclusions?

Reviewer #1: Partly

Reviewer #2: Partly

2. Has the statistical analysis been performed appropriately and rigorously? 

Reviewer #1: No

Reviewer #2: No

3. Have the authors made all data underlying the findings in their manuscript fully available?

Reviewer #1: No

Reviewer #2: Yes

4. Is the manuscript presented in an intelligible fashion and written in standard English?

Reviewer #1: Yes

Reviewer #2: Yes

5. Review Comments to the Author

Reviewer #1: Dear Author,

This article has an extensive study on developing a risk prediction nomogram to predict the late culture conversion among multi-drug resistant tuberculosis patients in North-West Ethiopia. This study has the novelty of developing the nomogram for the disease.

Critical Notes:

Abstract:

Line 39: Please use the term "binomial logistic regression" instead of "logit-based model." This might confuse the reader on many terms.

Line 46: Please explain the composition of the bootstrap parameters.

Line 52: keyword: "late culture conversion" should start with a capital letter?

Introduction:

Line 57: RR-TB ->

This abbreviation has not been previously defined (rifampicin-resistant TB)

Methodology:

Line 147–148: Did the authors consider checking the outliers and missing data?

Line 157: Please cite the reference for a p-value of 0.25.

Line 157: Please correct the "bellow" to "below." There are some more places in this article that need to be corrected.

Line 159–160: "The final streamlined......"How about the multicollinearity and interaction terms? Did the authors check on it?

Line 171: Please be consistent in using the p-value term "not just p".

Please use small capital letters for this "Decision Curve Analysis" (decision curve analysis).

Section 2.5.3, Please explain how the bootstrap procedure is employed. How many are in the bag and how many are out of it? What is the probability parameter used in this procedure?

Lines 182–183: The abbreviation should come after the definition.

Results:

Please be consistent with the use of percent and "%".

Section 3.5: Please use "variable" instead of "parameter".

Line 211: Why limit the candidate parameters to 14 only? Please justify it statistically.

Please report the results from the binomial logistic regression.

Please revise the decision on the selected variable/parameter for the nomogram.

Overall, the article needs to improve on the statistical analysis and presentation of the result.

Reviewer #2: 1. How many patients involved? 578 (under Results section) and the analysis was based on the 316 patients (It was not stated the final patients was 316 in the Results section)

2. What were the 14 variables involved as mentioned in "number of candidate parameters was limited to 14"

3. Please revise the results presentation.

4. Please justify this equation "Estimated risk of late culture conversion= 1/(1 + exp − (−0.323 + 0.63× BMI(<18.5) + 0.87 ..."

5. Please add the results of Binomial logistic regression.

6. Potential variables for late culture conversion prediction model development were considered 154 based on their easily obtainability, biologically plausible relationship with the outcome, and ease of interpretation in clinical practice. However, this statement was contradicting with "Those variables with p-value of 0.25 and bellow in univariable analysis were entered into multivariable analysis"

7. Please recheck this statement "The cut-off point identified by the first three methods was found to be 0.4562 and the one identified by SpEqualSe method was 0.4955." and Table 5.

6. PLOS authors have the option to publish the peer review history of their article (what does this mean?). If published, this will include your full peer review and any attached files.

Reviewer #1: **Yes: **Mohammad Nasir Abdullah

Reviewer #2: No

---

## [Author Response · Author response to Decision Letter 0]

11 Jul 2022

A rebuttal letter 

Journal name: PLOS ONE 

PONE-D-22-02882

Title: Development and validation of a nomogram for the prediction of late culture conversion among multi-drug resistant tuberculosis patients in North West Ethiopia: an application of Prediction modeling 

The Editor’s, reviewers’ comments, and point by point response of the authors are presented by the following table. For the sake of presenting the response clearly, we preferred the table form for this response. 

Editor’s comments Authors’ response

 Thank you editor, for your valuable comment. We have revised the manuscript for the journal’s style requirements, and the necessary corrections are made.

a) Did participants provide their written or verbal informed consent to participate in this study?

 Thank you Editor, for you comment. The data was secondary and collected by chart review. The IRB of university of Gondar has approved it and the necessary permission was taken from each institution where the study was conducted. We have written this in the revised manuscript. 

We will update your Data Availability statement to reflect the information you provide in your cover letter. Thank you editor, for your comments. 

There are no ethical or legal restrictions to sharing our data. Hence, as per your request, the minimal underlying data set is uploaded with the revised manuscript. 

 Thank you editor, for offering us your valuable comments. As per your suggestion, we have written the ethics statement in the method section of the revised manuscript.

5. Please include your tables as part of your main manuscript and remove the individual files. Please note that supplementary tables (should remain/ be uploaded) as separate "supporting information" files.

 We would like to give you our gratitude, for your kind and valuable comments. We have made the necessary correction and tables are included as part of the manuscript, and we have no supplementary tables to be uploaded. 

Comments from Reviewer #1 

This article has an extensive study on developing a risk prediction nomogram to predict the late culture conversion among multi-drug resistant tuberculosis patients in North-West Ethiopia. This study has the novelty of developing the nomogram for the disease.

Critical Notes:

Abstract:

Line 39: Please use the term "binomial logistic regression" instead of "logit-based model." This might confuse the reader on many terms.

Line 46: Please explain the composition of the bootstrap parameters.

Line 52: keyword: "late culture conversion" should start with a capital letter?

 Thank you reviewer, for your interesting and deep comments. As per your comments, the necessary correction is made in the abstract section of the manuscript, and indicated by the track changes. As per your suggestion, we have replaced “logit-based model” by binomial logistic regression. 

Introduction:

Line 57: RR-TB ->

This abbreviation has not been previously defined (rifampicin-resistant TB) Thank you reviewer for your valuable comment. Based on your comment, we have written the full word of the abbreviation at its first use in the revised manuscript. 

Methodology:

Line 147–148: Did the authors consider checking the outliers and missing data?

Line 157: Please cite the reference for a p-value of 0.25.

Line 157: Please correct the "bellow" to "below." There are some more places in this article that need to be corrected.

Line 159–160: "The final streamlined......"How about the multicollinearity and interaction terms? Did the authors check on it?

Line 171: Please be consistent in using the p-value term "not just p".

Please use small capital letters for this "Decision Curve Analysis" (decision curve analysis).

Sections 2.5.3, Please explain how the bootstrap procedure is employed. How many are in the bag and how many are out of it? What is the probability parameter used in this procedure?

Lines 182–183: The abbreviation should come after the definition.

 Thank you reviewer for these constructive comments.

-Yes, we have checked for outliers and missing data. We preferred to do multiple imputations instead of complete case analysis, for we don’t need to reduce the sample size. The technique of imputation and the list of variables with missing data are written in the method section of the revised manuscript, and also indicated with track changes. 

-Using pseudo linear regression, we have checked for multi-collinearity using VIF (variance inflation factor) and we found no multi-collinearity. 

-The bootstrapping procedure was made in R version 4.0.5 for the sake of internal validation. The random sampling was made by 10,000 reputations with replacement. The 95% confidence was set and, the output which contains the original coefficients, the bias and standard error(SE) was found. 

-Other comments which we haven’t responded here are also corrected in the revised manuscript and are shown with track changes. 

Results:

Please be consistent with the use of percent and "%".

Section 3.5: Please use "variable" instead of "parameter".

Line 211: Why limit the candidate parameters to 14 only? Please justify it statistically.

Please report the results from the binomial logistic regression.

Please revise the decision on the selected variable/parameter for the nomogram.

Overall, the article needs to improve on the statistical analysis and presentation of the result.

 Thank you dear reviewer for this important comment. 

-The revised manuscript is edited for your comments regarding consistency with the use of percent and “%”, variable instead of parameter. The binomial logistic regression result is also reported. 

- The variables are actually those which are reported in different literatures as prognostic determinants of late culture conversion. The number of events per parameters (EPP) has to be at least greater or equal to 10 to prevent the problem of over fitting in prediction modeling. 

Comments from Reviewer #2 

1. How many patients involved? 578 (under Results section) and the analysis was based on the 316 patients (It was not stated the final patients was 316 in the Results section)

 Thank you dear reviewer for your comment. The total patients on which the analysis done were 316. These were patients for whom culture conversion status was determined. As per your comment, we have stated it in the result section of the revised manuscript. 

2. What were the 14 variables involved as mentioned in "number of candidate parameters was limited to 14"

 Thank you reviewer for your valuable comments. The 14 variables involved were; sex, age, residence, treatment supporter, functional status at admission, registration group, HIV co-infection, flour quinolone resistance, baseline Body Mass Index (BMI), baseline anemia, sputum smear grade, radiological findings, regimen type and major adverse event.

-The above mentioned prognostic determinants are written in the method section of the revised manuscript under variables of the study sub-section.

3. Please revise the results presentation.

 Thank you reviewer for your valuable and kind comments. As per your suggestion, we have revised the results presentation in the revised manuscript.

4. Please justify this equation "Estimated risk of late culture conversion= 1/(1 + exp − (−0.323 + 0.63× BMI(<18.5) + 0.87 ..."

 Thank you dear for this comment too.

The equation is actually the probability equation of the binomial logit model. This is mathematics intensive and not feasible to calculate. Hence, the developed nomogram is a replace for this sophisticated equation to calculate the probability or risk of late culture conversion. The nomogram developed doesn’t require any calculator, for it is simple and user friendly graphical interface to calculate the individualized risk of patients for the outcome of interest. 

5. Please add the results of Binomial logistic regression.

 Thank you for your constructive comment. The result of binomial logistic regression is presented in table 3 of the revised manuscript.

6. Potential variables for late culture conversion prediction model development were considered based on their easily obtainability, biologically plausible relationship with the outcome, and ease of interpretation in clinical practice. However, this statement was contradicting with "Those variables with p-value of 0.25 and bellow in univariable analysis were entered into multivariable analysis"

 Thank you dear reviewer for this constructive comment. We preferred to use variables which can be ascertained early or soon after patient admission for the sake of early determination of risks for late culture conversion, and hence intervention without being late. So, the easily obtainability, biologically plausible relationship with the outcome, and ease of interpretation were considered right at selecting potential variables just before univariable analysis. In other word, the considerations mentioned were used to select candidate variables for univariable analysis. The stated p-value was used to select candidate variables for multivariable analysis. 

7. Please recheck this statement "The cut-off point identified by the first three methods was found to be 0.4562 and the one identified by SpEqualSe method was 0.4955." and Table 5. Thank you reviewer. The statement is rechecked and we have re-stated it in the revised manuscript.

---

## [Editor Report · Decision Letter 1]

28 Jul 2022

Development and validation of a nomogram for the prediction of late culture conversion among multi-drug resistant tuberculosis patients in North West Ethiopia: an application of Prediction modelling

PONE-D-22-02882R1

Dear Dr. Anley,

We’re pleased to inform you that your manuscript has been judged scientifically suitable for publication and will be formally accepted for publication once it meets all outstanding technical requirements.

Kind regards,

Aleksandra Barac

Academic Editor

PLOS ONE

---

## [Editor Report · Acceptance letter]

1 Aug 2022

PONE-D-22-02882R1 

Development and validation of a nomogram for the prediction of late culture conversion among multi-drug resistant tuberculosis patients in North West Ethiopia: an application of Prediction modelling 

Dear Dr. Anley:

I'm pleased to inform you that your manuscript has been deemed suitable for publication in PLOS ONE. Congratulations! Your manuscript is now with our production department. 

Kind regards, 

on behalf of

Dr. Aleksandra Barac 

Academic Editor

PLOS ONE